# Impact of a Single-Tube PCR Assay for the Detection of *Haemophilus influenzae* Serotypes a, c, d, e and f on the Epidemiological Surveillance in Greece

**DOI:** 10.3390/microorganisms10071367

**Published:** 2022-07-07

**Authors:** Athanasia Xirogianni, Theano Georgakopoulou, Vassileios Patsourakos, Ioanna Magaziotou, Anastasia Papandreou, Stelmos Simantirakis, Georgina Tzanakaki

**Affiliations:** 1National Meningitis Reference Laboratory, Department of Public Health Policy, School of Public Health, University of West Attica, 11521 Athens, Greece; axirogianni@uniwa.gr (A.X.); vapa1917@gmail.com (V.P.); npapandreou@uniwa.gr (A.P.); ssimantirakis@uniwa.gr (S.S.); 2Department of Vaccine Preventable Diseases, National Public Health Organization (NPHO), 15123 Athens, Greece; t.georgakopoulou@eody.gov.gr (T.G.); i.magaziotou@eody.gov.gr (I.M.)

**Keywords:** multiplex PCR, *H. influenzae* serotypes, *H. influenzae* epidemiology surveillance

## Abstract

Background: The decrease in the rate of meningitis due to *Haemophilus influenzae* type b after vaccine introduction and a possible change in epidemiology of *H. influenzae* disease highlights the need for continuous serotype surveillance. Methods: A single-tube multiplex PCR assay for serotyping of *H. influenzae* was developed and deployed. Results: During 2003–2020, 108 meningitis cases due to *H. influenzae* were notified; 86 (80%) were confirmed and serotyped by molecular methods. The overall specificity and sensitivity of the assay were estimated (100% PPV and NPV respectively). The overall mean annual reported incidence for *H. influenzae* was 0.02, while for Hib and non-b meningitis equaled 0.02 and 0.03 per 100 000, respectively. Analysis by age group revealed that *H. influenzae* peaks in toddlers and children 0–4 years and in adults >45 years old. Among the serotyped cases, 39.8% were identified as Hib, 46.3% as NTHi, and 0.9% and 2.8% as serotypes a (Hia) and f (Hif)) respectively. Conclusions: Low incidence due to Hib was observed while non-typeable *H. influenzae* (NTHi) and serotypes Hia and Hif seem to emerge. The application of the current assay discloses the ongoing change of invasive *H. influenzae* disease trends during the Hib post-vaccine era.

## 1. Introduction

*Haemophilus influenzae* is a Gram-negative aerobic or facultative anaerobic coccobacillus that is differentiated according to its capsular polysaccharide composition into six serotypes (a–f) and noncapsulated, non-typeable *H. influenzae* (NTHi) strains [1]. Based on the capsule, encapsulated strains are more virulent -especially the *H. influenzae* serotype b (Hib) [2] implicated in the cause of severe invasive disease, such as, meningitis, septicemia and pneumonia [3]. Nevertheless, NTHi strains can also cause invasive disease, as well as non-invasive infections of upper respiratory tract [4,5].

Prior to the vaccine introduction in early 1990s, Hib caused more than 95% of invasive disease, especially in children less than five years of age [6], whereas non-type b *H. influenzae* caused opportunistic infections mainly among elderly persons with predisposing conditions, such as respiratory disease or immunosuppression [7,8,9]. Following Hib vaccination, several studies have shown a significant decrease in Hib incidence, as well as in carriage rates worldwide, with simultaneous increase in other serotypes (mainly a, e, f), and NTHi [10,11,12,13].

Almost three decades after vaccine introduction, great concern still exists on the long-term effectiveness of the Hib immunization programs. The emerging possibility of serotype replacement under vaccination pressure and the change in epidemiology of *H. influenzae* disease highlighted the need for continuous serotype surveillance. Thus, capsule identification is essential and particularly valuable to public health response and intervention.

Traditionally, *H. influenzae* serotypes are identified by slide agglutination tests using specific antisera. However, several PCR assays or similar methods for the detection and identification of *H. influenzae* serotypes have been developed during the last 20 years [14,15,16,17]. Additionally, serotypes can also be identified by the application of Whole Genome Sequencing [18]. However, all the above assays require culture methods and isolation of the bacterial strains.

As non-culture diagnosis covers about 70% of the *H. influenzae* cases in Greece (current data), the aim of the present study is to evaluate the development of a single-tube multiplex PCR (mPCR) assay for the simultaneous detection of *H. influenzae* serotypes a, c, d, e and f directly in clinical samples (whole blood, cerebrospinal fluid (CSF) as a tool for rapid non-culture diagnosis in order to enable continuous surveillance of *H. influenzae* disease in Greece in the post-vaccine era.

## 2. Materials and Methods

### 2.1. Sample Source

A total of 108 menigitis cases due to *H. influenzae* were notified to the National Public Health Organization (NPHO) through the mandatory notification system during the period 2003–2020. Of those, samples from 86 (79.6%) cases were sent to the National Meningitis Reference Laboratory (NMRL) for further identification. Those included 113 samples of whole blood (*n* = 4), CSF (*n* = 86) and bacterial isolates (*n* = 23), as for some cases more than one sample were analyzed. Ethical patient consent was waved due to depersonalized data from notifiable disease registry and laboratory findings were used.

### 2.2. Identification of H. Influenzae Serotype in Bacterial Isolates

All isolates were identified by conventional culture (chocolate agar culture) and biochemical methods (RapID NH SYSTEM, R8311001, Remel Europe Ltd., Dartford, UK). Serotyping was carried out by slide agglutination with capsular typing antisera (Remel Europe Ltd., Dartford, UK).

### 2.3. DNA Isolation

DNA isolation from clinical samples was carried out using the MagCore^®^ Genomic DNA Whole Blood Kit (MagCore HF 16 nucleic acid extraction system, RBC Bioscience, New Taipei City, Taiwan). For *H. influenzae* isolates, DNA extraction was carried out from 24 h chocolate agar culture [19]. In brief, bacterial colonies were suspended in 500 μL sterile double distilled H_2_O, vortexed, boiled for 15 min and centrifuged at 20,000× *g* for 12 min. The supernatant was retained, and the DNA concentration was estimated spectrophotometrically.

### 2.4. Identification of H. influenzae in Clinical Samples

For culture negative suspected samples, the application of two mPCR assays, for Hib and *H. influenzae* were deployed as previously described [19,20]. In brief, the first mPCR assay targeted the leading infectious agents of bacterial meningitis accounting for 70–80% such as *Neisseria meningitidis*, *Streptococcus pneumoniae* and *Haemophilus influenzae* type b, while the second mPCR assay targeted the microorganisms causing 20–30% bacterial meningitis such as *H. influenzae* (non-b) *Pseudomonas aeruginosa*, *Staphylococcus aureus* and *Streptococcus* spp.

### 2.5. Development of the Multiplex PCR Assay for the Identification of H. influenzae Serotypes

#### 2.5.1. PCR Primers

For the simultaneous detection of the *H. influenzae* serotypes a, c, d, e and f, a mPCR assay was designed with specific primers for each serotype, respectively, as described previously [15,21,22] (Table 1).

#### 2.5.2. Amplification Protocol

Amplification reactions contained 0.5 μM of each primer for serotype a (VBC, Hamburg, Germany), 0.2 μM of each primer for serotypes c, d, f and 0.1 μM of each primer for serotype e, 0.8 mM dNTPs (New England Biolabs, Ipswich, MA, USA), 0.7 U Phusion High-Fidelity DNA Polymerase (New England Biolabs), 1.3× reaction buffer GC and 3 μL of DNA template in a total volume 25 μL.

Polymerase chain reaction conditions were the following: 98 °C for 30 s; 98 °C for 5 s, 62 °C for 10 s and 72 °C for 50 s (10 cycles); 98 °C for 5 s, 60 °C for 10 s and 72 °C for 55 s (10 cycles); 98 °C for 5 s, 59 °C for 10 s and 72 °C for 55 s (10 cycles) and 98 °C for 5 s, 57 °C for 10 s and 72 °C for 55 s (5 cycles). The final extension step was at 72 °C for 2 min (Piko Thermocycler, Thermo Fisher Scientific Inc., Waltham, MA, USA).

DNA positive controls obtained from for serotypes a, c, d, e and f reference strains (ATCC 9006, ATCC 9007, ATCC 9008, ATCC 8142, ATCC 9833, 1 ng of each DNA) as well as negative controls were included in each PCR assay.

Further, gel electrophoresis was carried out in 5 μL of the PCR product stained with with GelRed loading buffer (6X Gel loading dye, Biotium, Fremont, CA, USA) in 2% (*w*/*v*) agarose gel (Nippon Genetics, Tokyo, Japan) and visualized under ultraviolet fluorescence light.

### 2.6. Sensitivity and Specificity Assessment

Sensitivity was tested with DNA extracted from 73 culture confirmed samples, including 63 strains isolated from patients with invasive (*n* = 23) and non-invasive (*n* = 40) disease and 10 External Quality Assurance (EQA) control strains (UK-NEQUAS, Sheffield, UK). Specifically, all isolates were initially serotyped by conventional methods (seroagglutination test). For the PCR assay’s detection limits, serial dilutions of spectrophotometrically quantified DNA (0.5–0.001 ng) of the positive control DNA from each serotype were amplified. As the proposed mPCR assay was applied in clinical samples with a large amount of human genomic DNA and smaller quantities of bacterial DNA, Phusion Taq polymerase (New England Biolabs, Ipswich, MA, USA) was used in combination with two additional annealing steps in order to achieve high specificity and sensitivity [23].

## 3. Results

### 3.1. Assessment of the Multiplex PCR Assay Performance

#### 3.1.1. Sensitivity and Specificity

All *H. influenzae* strains for which the seroagglutination tests did not identify any of the assigned serotypes a, c, d, e and f, were also negative by the current mPCR (Hib (*n* = 11) and NTHi (*n* = 59)). In contrast, the mPCR assay correctly identified serotype f (*n* = 2) and serotype a (*n* = 1). The overall sensitivity and specificity of the multiplex PCR assay was estimated to 100%_,_ with 100% PPV and NPV (Positive and Negative Predicted Value). Further, by the use of Phusion Taq polymerase in combination with two additional annealing steps, a low quantity of 0.001 ng was detected.

#### 3.1.2. Identification of *H. influenzae* in Clinical Samples and Isolates from Patients with Meningitis

Among the 113 tested clinical samples, 44 samples [whole blood (*n* = 4), CSF (*n* = 33), isolates (*n* = 7)] identified as *H. influenzae* type b (Hib) were negative by the application of the current mPCR assay. Further, among the 63 samples [CSF (*n* = 49), isolates (*n* = 14)] identified as non-type b *H. influenzae*, were identified as NTHi, as no serotype was identified by the current assay. Finally, six samples [CSF (*n* = 4), isolates (*n* = 2)] were positive by the current mPCR assay, one sample successfully identified serotype a (Hia) and five serotype f (Hinf).

### 3.2. Application to the Epidemiological Surveillance (2003–2020)

#### 3.2.1. Serotype Distribution by Clinical Presentation

Overall, 108 meningitis cases due to *H. influenzae* were notified during the period 2003–2020 of which nine were presented also with septicemia. Among those, 39.8% (43/108) were identified as Hib (either by mPCR assay (*n* = 33) or by conventional methods (*n* = 10), serotypes a (Hia) and f (Hif)) were successfully identified in 0.9% (1/108) and 2.8% (3/108), respectively. Further, 46.3% (50/108) of the cases were NTHi, while samples from 11 notified cases (10.2%) were not sent to NMRL for further typing.

#### 3.2.2. Incidence

Meningitis is under mandatory surveillance in Greece and thus the overall mean annual reported incidence of *H. influenzae* meningitis was 0.02/100,000 population during the study period (2003–2020). The mean annual reported incidence of Hib and non-b meningitis equaled 0.02 and 0.03, respectively.

Hib incidence remained low and below 0.08/100,000 population during 2003–2009 while a further decrease was observed since 2010 with a simultaneous increase of non b *H. influenzae* (Figure 1). The annual incidence for Hib meningitis peaked in 2004 (0.07/100,000 population), while a peak of similar magnitude for non-b meningitis incidence was recorded in 2013 (0.07/100,000 population).

#### 3.2.3. Serotype Distribution by Age Group and Time Period

Analysis of meningitis cases revealed that *H. influenzae* peaks in toddlers and children 0–4 years of age (39.8%; 43/108) and in adults > 45 years old (42.6%; 46/108) (Figure 2).

Among the age group 0–4 years, of 41 cases serotyped, the majority (70.7%; 29/41) were due to Hib followed by NTHi (24.4%; 10/41), while a low rate (4.9%; 2/41) of serotype f was observed. Further analysis by age and serotype revealed that most of the Hib cases in this age group were identified in infants less than a year old (62.1%, 18/29). Among those, 77.8% (15/18) were either not vaccinated (three due to age non-eligibility i.e., <2 months old) or not fully vaccinated (one case). In addition, a significant proportion of children < 1 and 1–4 years of age (50.0% and 45.5% respectively) belonged to subpopulation groups such as Roma, immigrants and refugees. In children 5–14 years old, NTHi predominated in 71.4% (5/7) of cases while the rest (2/7) were Hib.

As regards adults > 25 years of age, serotyping was performed in 49 of the 58 cases. In total, 71.4% (35/49) were NTHi followed by Hib (24.5%; 12/49) while one case of serotype a and one of serotype f were identified in a 45-year-old male and a 61 year old female respectively (Figure 2).

For analysis purposes, the data were split in two study periods (2003–2009 and 2010–2020) (Figure 3a,b). The number of *H. influenzae* cases in the age group of 0–4 remained similar (22 vs. 21) during the study periods. However, a 2.6-fold decrease was observed for Hib during 2010–2020 (*n* = 8) compared to the first time period (2003–2009) (*n* = 21) as Hib seems to be replaced mainly by NTHi (47.6%; 10/21) and serotype f (*n* = 2) during the second period (2010–2020).

In contrast, for adults > 25 years of age, an increase in recorded cases was evident in the second time period (40 vs. 18). Comparing serotyped cases, the proportion of cases due to Hib considerably decreased during the second time period (from 41.2% to 15.6%) while a substantial increase was recorded in the proportion of NTHi cases (from 58.8% to 78.1%), depicting also the difference in the number of cases between the two time periods (10 cases in 2003–2009 vs. 25 cases in 2010–2020).

Low numbers of Hib cases were observed in the age group of 5–19 years during both time periods.

## 4. Discussion

The study presents the development of a new single-tube multiplex PCR assay for the simultaneous detection of five *H. influenzae* serotypes (a, c, d, e, f) directly in clinical samples, providing a relatively inexpensive, rapid and reliable method for the non-culture serotype identification of invasive *H. influenzae* disease. The developed assay is particularly valuable for the confirmation of *H. influenzae* in cases where initiation of early antibiotic treatment prevents detection by culture. Moreover, in comparison with other serotyping methods [14,15,16,17,18], it is well applied on clinical samples such as whole blood further to CSF samples and bacterial isolates.

Previous relative studies on the *H. influenzae* type b epidemiology during the pre- and post-vaccination era in Greece, conducted in the paediatric population in the area of Athens and Crete, reported an incidence of 12.0/100,000 population [24], with meningitis accounting for 69% of the cases. Hib vaccination was introduced in Greece in 1992 and led to a limitation of meningitis cases [25,26,27]. During the same period, a study by Kofteridis et al. reported high percentage (58%) of invasive disease of lower respiratory tract due to NTHi [28]. In the present study, a 17-year period collection of *H. influenzae* meningitis cases from throughout the country is presented, including all notified cases to the National Public Health Organization, of which the 86 cases (via National Meningitis Reference Laboratory) were laboratory confirmed mainly by PCR assays and serotyped with the presented mPCR assay.

The mean annual reported incidence of *H. influenzae* meningitis is 0.02 per 100,000 population with an average of 6 cases recorded annually. Although a decrease in Hib cases was observed especially after vaccine implementation, there was an increase of non-b *H. Influenzae* cases. During the last three years (2018–2020), no cases of Hib meningitis have been notified resulting to zero Hib incidence. The peak annual incidence for Hib meningitis was recorded in 2004 with a lower one in 2009 while non-b incidence peaked during the more recent years, in 2013 and 2015. Trends of replacement seem to have started a decade ago and have gradually established, especially during the past five years with a reduction of notified Hib meningitis cases.

Hib immunization in Greece can be considered as well established. According to the National Vaccination Program, immunization against Hib starts in infants of two months of age and consists of three initial doses with an interval of two months and a booster dose at the age of 15–18 months (three + one doses). As recorded in a national cross-sectional vaccination coverage study in preschool children attending nurseries-kindergartens, the vaccination coverage with one, two, three and four doses is very high (100%, 99.9%, 99.4% and 95.2%, respectively) [29]. The same study showed equally high rates for the completeness of vaccination with three doses by 12 months for both the general population and immigrant children (95.3% vs. 94.3%). However, a delay has been recorded in the administration of the four doses by 24 months with a significant difference between the general population and immigrants (84.9, 95% CI = 83.4–86.3 vs. 78.6%, 95 CI = 71.0–84.6; *p* = 0.042) [29]. Furthemore, a report for the European Commission in 2018 showed that the confidence in vaccines in Greece is higher than the EU average [30].

Our results reveal the impact of vaccination in infants less than one year of age, as the majority of cases identified as Hib were not vaccinated and some partly vaccinated. As a result of the successful implementation of Hib vaccination in younger ages, fewer Hib cases are being notified in adults. So far, no vaccine-failure cases have been reported in Greece, although recent relative studies reveal vaccine failure in Europe [31,32,33].

Overall, our results are in agreement with those of other European countries. Hib vaccination seems to have a great impact in the decrease *of H. influenzae* invasive disease [34]. Meanwhile, an increase on invasive disease due to NTHi and serotypes f, a and e is observed in several European countries such as Italy, France, Ireland and Germany in infants and the elderly (>60 years old) [33,35,36,37], whereas in countries such as the UK, Germany and Denmark, a slight increase of serotypes e and f, is evident in adults with underlying conditions [12,37,38,39,40]. In the last decade, *H. influenzae* serotype e was also reported as a causative pathogen of meningitis in Europe following serotypes f and a, as shown recently in a case report published from Turkey [41].

## 5. Conclusions

In countries with well-established vaccination programs and sustained high vaccination coverage, Hib meningitis has greatly decreased. However, emergence of NTHi, Hia, Hie, and Hif is apparent in recent years. The changing epidemiology of *H. influenzae* highlights the need for continuous surveillance and serotype monitoring. The application of the proposed methodology for serotyping directly to culture negative, PCR positive clinical samples is relatively inexpensive, rapid and reliable. Moreover, the proposed mPCR assay can ameliorate *H. influenzae* serotype surveillance, which is crucial for the evaluation of the overall impact of vaccination programs and for designing potential future vaccination strategies.

## Figures and Tables

**Figure 1 microorganisms-10-01367-f001:**
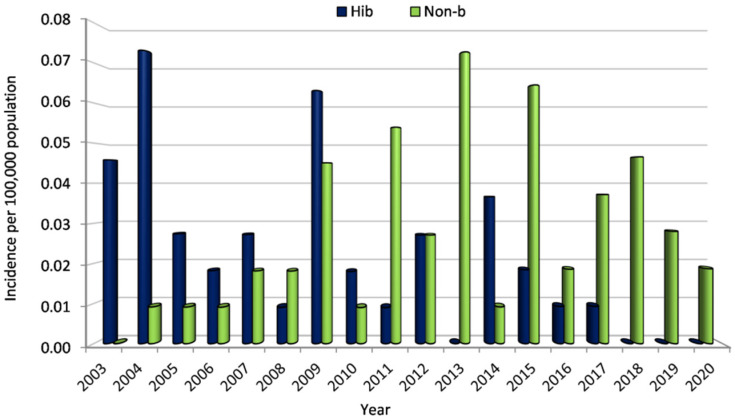
Incidence of *H. influenzae* b and non-b meningitis (2003–2020).

**Figure 2 microorganisms-10-01367-f002:**
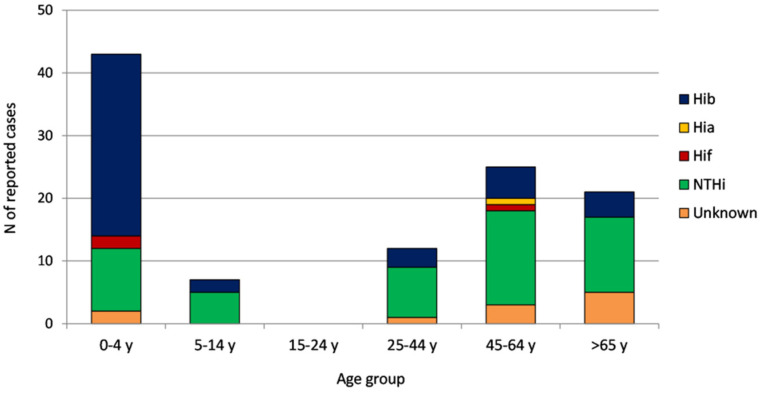
Serotype distribution of meningitis cases due to *H. influenzae* by age group.

**Figure 3 microorganisms-10-01367-f003:**
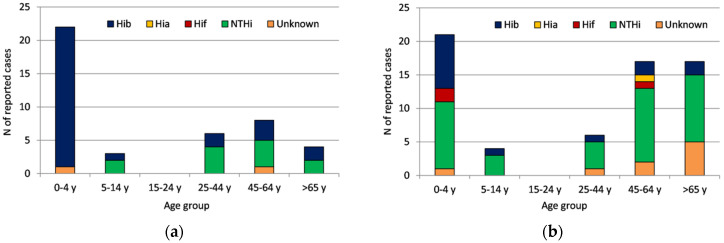
*H. influenzae* serotype distribution by age group during two study periods (**a**) 2003–2009 and (**b**) 2010–2020.

**Table 1 microorganisms-10-01367-t001:** Oligonucleotide primers used for the identification of *H. influenzae* serotypes.

Serotype	Primers	Sequences (5′-3′)	Amplicon Size (bp)	Publication
a	haf	ATCTTACAACTTAGCGAATAC	1180	modified [15] (6 nucleotides from the 5′ edge have been deleted)
	ha2	GAATATGACCTGATCTTCTG		[21]
c	hc2	CAGAGGCAAGCTATTAGTGA	200	[21]
	hc3	TGGCAGCGTAAATATCCTAA		
d	hd1	TGATGACCGATACAACCTGT	150	
	hd2	TCCACTCTTCAAACCATTCT		
e	TTL 63	GAGCAATTCCATCGTAGTAAC	592	[22]
	TTL 64	TTCTTCATATCCTCGCAATTG		
f	hf1	GCTACTATCAAGTCCAAATC	456	[21]
	hf2	CGCAATTATGGAAGAAAGCT		

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
