# Peer review of "Impact of a Single-Tube PCR Assay for the Detection of Haemophilus influenzae Serotypes a, c, d, e and f on the Epidemiological Surveillance in Greece"

_microorganisms, 2022, doi:10.3390/microorganisms10071367_

Round 1
Reviewer 1 Report
Dear authors,
this is a very interesting piece of work. A well designed and well written study with important findings. apart from some spelling mistakes such as the one mentioned below
in row 123 'did not identified' should be corrected to 'did not identify'
this is a work worth publishing.
Author Response
Dear authors,
this is a very interesting piece of work. A well designed and well written study with important findings. apart from some spelling mistakes such as the one mentioned below in row 123 'did not identified' should be corrected to 'did not identify'. this is a work worth publishing.
Authors’ response: The authors would like to thank the reviewer for his/her time to revise the manuscript as well as for the kind comments. The spelling mistake has been corrected (now line 157)
Reviewer 2 Report
The source of testing specimen is confused. As the title indicates ....... H. influenzae serotypes a, c, d, e & f directly in clinical samples...... (directly from the clinical specimen?); however, in the section of materials and methods (line 84), …………….. for H. influenzae isolates, DNA extraction was carried out from 24 h chocolate agar culture (from culture specimen?).
This question should be clarified.
Moreover, the description of materials and methods can be improved to be clearer for the readers.
Author Response
Response to Reviewer -2
- The source of testing specimen is confused. As the title indicates ....... influenzae serotypes a, c, d, e & f directly in clinical samples...... (directly from the clinical specimen?); however, in the section of materials and methods (line 84), ……………..for H. influenzae isolates, DNA extraction was carried out from 24 h chocolate agar culture (from culture specimen?).
Authors’ response: The authors would like to thank the reviewer for the comment. Although the proportion of clinical samples was significantly higher compared to the culture positive samples, the title has been changed by deleting the wording “directly in clinical samples”
This question should be clarified.
- Moreover, the description of materials and methods can be improved to be clearer for the readers.
Authors’ response: Thank you for this comment. We believe that the material and method section has been now improved by :
- Specifying clearly the number of notified cases in relation to the cases and samples received at the NMRL for further identification (now lines 68-77)
- By briefly describing the DNA isolation from isolates (now lines 93-96)
- By briefly describing the 2 mPCR assays and the target microorganisms (as described in the Refs 19,20) (now lines 99-103)
- By adding in a new sub heading for sensitivity and specificity assessment (now lines 138-149)
Reviewer 3 Report
Xirogianni A, et al investigated the detection of H influenzae serotypes directly in clinical isolates. I accept the importance of the study, but some serious comments have emerged. The advantage of the research was the data on serotype’s distribution for a very long period between 2003-2020
1. Why did the authors decide to exclude Hib detection from mPCR? Why they use a separate PCR reaction? Why did they name this reaction mPCR – which genes besides the capsule b gene are involved?
2. The validation of the mPCR detection reaction, ssessment of specificity and sensitivity requires more isolates – the use of only 5 isolates with a specific serotype is too small . Line 126 the authors state that they use 4 isolates Hib, one Hif and five NTHi. In my opinion at least 10 isolates per serotype should be included.
In addition, there is a discrepancy as the authors stated in material and method section that they used 40 isolates NTHi and 10 EQA control strains to evaluate specificity and sensitivity. How the authors will explain this discrepancy.
Line 87.89 – please provide more details for mPCR
Additional validation of the mPCR single tube reaction would be good to made – if the authors isolated 23 bacterial isolates from csf the authors can compare what serotypes were given by conventional methods and what serotypes of mPCRs. This will be useful as the authors report that this mPCR is used directly for clinical specimens
Author Response
Response to Reviewer -3
- Why did the authors decide to exclude Hib detection from mPCR? Why they use a separate PCR reaction? Why did they name this reaction mPCR – which genes besides the capsule b gene are involved?
Authors’ response: Thank you for this comment. As Neisseria meningitidis, Streptococcus pneumoniae and Haemophilus influenzae type b, are the leading cause of bacterial meningitis accounting of 70-80% of the cases, a first mPCR was developed to target the aforementioned bacteria in 2004 ((Tzanakaki et al 2005 REF 19). However, as 20-30% of the samples received remained negative, we aimed to develop the second mPCR in 2008 targeted the microorganisms H. influenzae (non-b) Pseudomonas aeruginosa, Staphylococcus aureus and Streptococcus spp. In order was to overcome the aforementioned problem and to further identify the microorganism causing bacterial meningitis (REF 20 Xirogianni et al 2009). This has been added also in the text (lines 99-103)
- The validation of the mPCR detection reaction, ssessment of specificity and sensitivity requires more isolates– the use of only 5 isolates with a specific serotype is too small . Line 126 the authors state that they use 4 isolates Hib, one Hif and five NTHi. In my opinion at least 10 isolates per serotype should be included.
Authors’ response: We thank you for this comment. In fact, -as also clarified in the text (lines 138-148) a total of 73 isolates were tested by seroagglutination test and not only the EQA isolates as stated. However, as shown also in the epidemiological results, the serotypes a,c,d,e and f are rare not only in Greece but also in many European countries. Hence, there was no possibility for testing more than the already tested.
- In addition, there is a discrepancy as the authors stated in material and method section that they used 40 isolates NTHi and 10 EQA control strains to evaluate specificity and sensitivity. How the authors will explain this discrepancy.
Authors’ response: This discrepancy is clarified in the previous comment and also now clarified in the text (lines 138-148).
Line 87.89 – please provide more details for mPCR
Authors’ response: Both published mPCR assays (refs 19,20) are briefly described in lines 99-103
- Additional validation of the mPCR single tube reaction would be good to made – if the authors isolated 23 bacterial isolates from csf the authors can compare what serotypes were given by conventional methods and what serotypes of mPCRs. This will be useful as the authors report that this mPCR is used directly for clinical specimens
Authors’ response: Thank you for the comment. In all 22 of 23 cultured confirmed cases, CSF samples were also sent prior to the isolated strains. This was a very good indicator as the seroagglutination results confirmed the non-culture serotyping proposed mPCR assay.
Round 2
Reviewer 3 Report
The authors responded satisfactorily to the comments.